# Prediction of virus-host associations using protein language models and multiple instance learning

**Dan Liu**[1], **Francesca Young**[1,2], **Kieran D. Lamb**[1], **David L. Robertson**[1]*, **Ke Yuan**[2,3,4]*

**1** MRC−University of Glasgow Centre for Virus Research, Glasgow, United Kingdom, **2** School of Computing Science, University of Glasgow, Glasgow, United Kingdom, **3** School of Cancer Sciences, University of Glasgow, Glasgow, United Kingdom, **4** Cancer Research UK Scotland Institute, Glasgow, United Kingdom

* david.l.robertson@glasgow.ac.uk (DLR); ke.yuan@glasgow.ac.uk (KY)

**Data Availability Statement:** Binary and multiclass data are collected from https://www.genome.jp/virushostdb/. Code and trained models are available on https://github.com/liudan111/EvoMIL.

## Abstract

Predicting virus-host associations is essential to determine the specific host species that viruses interact with, and discover if new viruses infect humans and animals. Currently, the host of the majority of viruses is unknown, particularly in microbiomes. To address this challenge, we introduce EvoMIL, a deep learning method that predicts the host species for viruses from viral sequences only. It also identifies important viral proteins that significantly contribute to host prediction. The method combines a pre-trained large protein language model (ESM) and attention-based multiple instance learning to allow protein-orientated predictions. Our results show that protein embeddings capture stronger predictive signals than sequence composition features, including amino acids, physiochemical properties, and DNA k-mers. In multi-host prediction tasks, EvoMIL achieves median F1 score improvements of 10.8%, 16.2%, and 4.9% in prokaryotic hosts, and 1.7%, 6.6% and 11.5% in eukaryotic hosts. EvoMIL binary classifiers achieve impressive AUC over 0.95 for all prokaryotic hosts and range from roughly 0.8 to 0.9 for eukaryotic hosts. Furthermore, EvoMIL identifies important proteins in the prediction task, capturing key functions involved in virus-host specificity.

## Author summary

Being able to predict which viruses can infect which host species, and identifying the specific proteins that are involved in these interactions, are fundamental tasks in virology. Traditional methods for predicting these interactions rely on identifying common features among proteins, overlooking the structure of the protein "language" encoded in individual proteins. We have developed a novel method that combines a protein language model and multiple instance learning to allow host prediction directly from protein sequences, without the need to extract features manually. This method significantly improved prediction accuracy and revealed key proteins involved in virus-host interactions.

**Funding:** DL is funded by European Union's Horizon 2020 research and innovation program, under the Marie Skłodowska-Curie Actions Innovative Training Networks grant agreement no. 955974 (VIROINF). The authors also acknowledge support from the following grants: the Medical Research Council (MRC, MC_UU_12014/12, MC_UU_00034/5, MR/V01157X/1) to DLR, a Doctoral Training Programme in Precision Medicine studentship for KDL, MR/N013166/1, the Biotechnology and Biological Sciences Research Council (BBSRC, BB/V016067/1) to DLR, FY and KY, and Engineering and Physical Sciences Research Council (EPSRC, EP/R018634/1) to KY. The funders had no role in study design, data collection and analysis, decision to publish, or preparation of the manuscript.

**Competing interests:** The authors have declared that no competing interests exist.

## Introduction

Advances in sequencing technologies, particularly metagenomics, have resulted in the identification of many new viruses. However, more than 90% of the virus sequences held in publicly available databases are not annotated with any host information [1]. Currently, there are no high-throughput experimental methods that can definitively assign a host to these uncultivated viruses. With a growing number of viruses being discovered, relying only on experiments to identify virus-host associations is a limiting step in this important challenge.

A number of computational approaches have been developed to predict unknown virus-host species associations. The coevolution of a virus and its host leave signals in virus genomes arising from the virus-host interaction. These signals have been exploited for in silico prediction of virus-host associations from virus genomes alone and fall into two broad types: 1) alignment-based approaches that search for homology such as prophage [2], CRISPR-cas spacers [3, 4]; 2) Alignment-free methods that use features such as k-mer composition, codon usage, CpG content etc. to measure the similarity between viral and host sequences or to other viruses with a known host [5]. To date, no computational approaches consider the structure of proteins from viruses for host species prediction purposes.

Here, we present a virus-host prediction model combining protein language models (PLMs) and multiple instance learning (MIL). Transformers are self-supervised deep learning models [6] that learn the relationships among words within a sentence, and now dominate the field of natural language processing. More recently, the same architecture has been applied in biology, where words are replaced by amino acids and sentences by protein sequences. These transformer-based protein language models generate protein embeddings that encode structural features inferred from amino acid sequences based on large-scale protein databases [7]. Protein language models are trained on publicly available protein sequence archives and learn biological information from physiochemical properties of the individual amino acids to structural and functional information about proteins. Multiple instance learning (MIL) is a form of supervised learning that was developed for image processing tasks [8]. Instead of using individually labelled instances for classification, multiple instances are arranged together in a bag with a single label and classified together. We use attention-based MIL [9], which has the additional advantage of weighting instances in a bag, thereby indicating the importance of each instance in prediction.

The combination of the two approaches is particularly suited for virus-host prediction, as virus proteins collectively contribute to the association with a host. Instead of relying on predefined features, protein language models provide automatically learned features, free from design biases and limitations of the previous approaches. The ability to measure similarity and differences between protein sequences further boosts prediction performance through multiple instance learning, where viral proteins enabling interaction with hosts are highlighted through unbiased weight estimation.

In this paper, we introduce EvoMIL a method for predicting virus-host associations by combining the (Evo)lutionary Scale Modeling with (M)multiple (I)instance (L)earning, Fig 1. EvoMIL uses the model ESM-1b [7] to transform viral protein sequences into embeddings (i.e., numerical vectors) that are then used as features for virus-host classification. Multiple-instance learning allows us to consider each virus as a "bag" of proteins. We demonstrate that the embeddings capture the host signal from the viral sequences achieving high prediction scores at the species level of both prokaryotic and eukaryotic hosts. Furthermore, attention-based MIL enables us to identify which proteins are highly important in driving prediction and by implication are important to virus-host specificity.

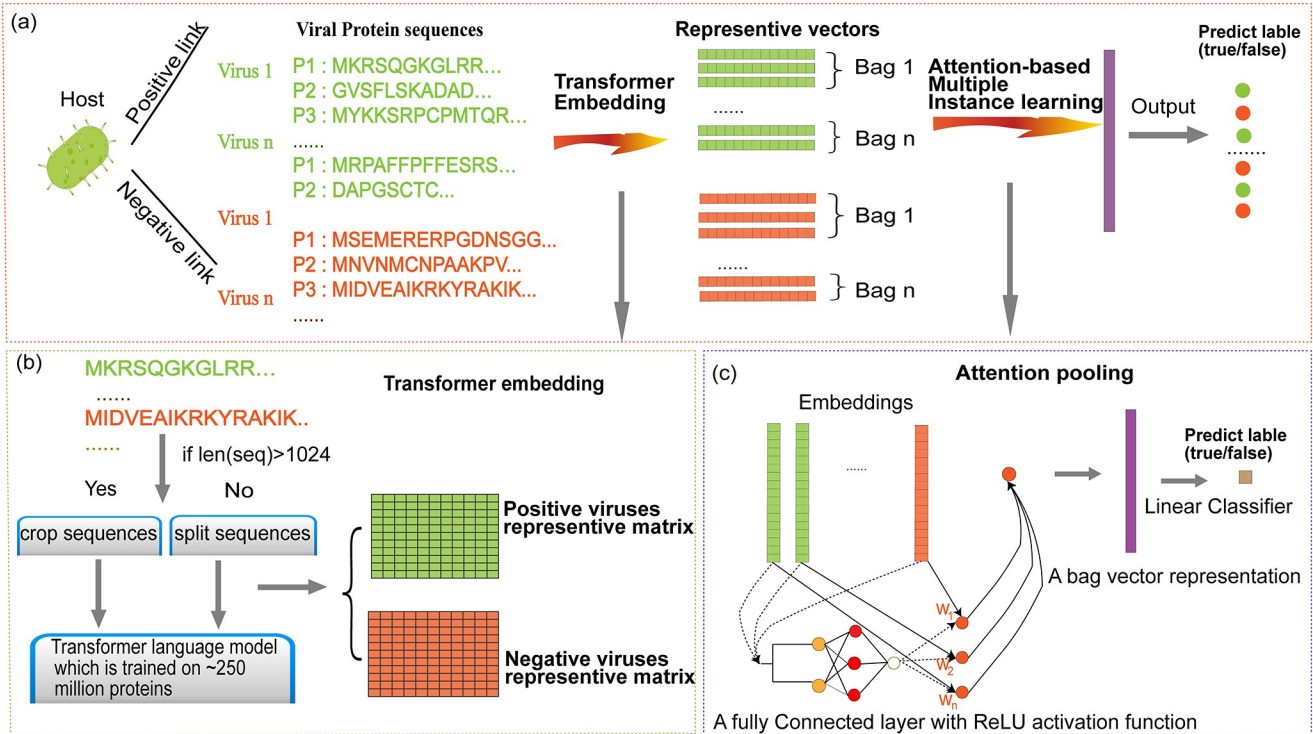

**Fig 1. A diagrammatic representation of the EvoMIL method.** (A) Protein sequences of viruses and virus-host associations are collected from the VHDB [10]. For each host, we collect the same number of positive and negative viruses, and then embeddings of protein sequences from viruses are obtained by the pre-trained transformer model [7], which are features for host predictions based on attention-based MIL; (B) Protein sequences of viruses are split to sub-sequences, which are used as input to the pre-trained transformer model to obtain the corresponding embeddings; (C) There is a host label for a set of protein sequences on each virus, and attention-based MIL is applied to train the model for each host dataset by protein embeddings of viruses. Finally, we predict the host label for each virus and assign an instance weight that represents the importance of each protein for the virus.

## Results

### Dataset for predicting virus-host association

Balanced binary datasets were generated from known virus-host associations documented in the Virus-Host database (VHDB) [10] for all hosts with a minimum threshold number of associations. These datasets consist of either all prokaryotic or all eukaryotic viruses. 'Positive' viruses are those that are reported to be associated with the given host species. A matching number of 'negative' viruses are randomly sampled from all other prokaryotic or eukaryotic viruses. The prokaryote datasets consist of nearly all dsDNA (double-stranded DNA) viruses which have 45 to 212 proteins coded in their genomes (S1 Table), while the eukaryotic datasets include many RNA viruses that contain fewer proteins, ranging from 2 to 23 protein sequences (S2 Table). The performance of MIL improves with higher numbers of instances in each bag, therefore we need to increase the threshold for the number of viruses in the eukaryotic training datasets to achieve similar performance with MIL. Accordingly, we set a threshold for minimum positive dataset size to 50 and 125 viruses for constructing prokaryotic and eukaryotic binary datasets, respectively. The aim of setting the threshold is to generate a sufficient number of training samples for MIL training on prokaryotic and eukaryotic hosts, respectively. Finally, we generated 15 prokaryotic host datasets and 5 eukaryotic host datasets for the binary classification tasks.

To evaluate the performance of binary models, we created a balanced set of positive and negative samples. For negative samples, we used two different strategies to sample the negative viruses from those with no known association with each host identified above to create balanced binary datasets. Given that the actual association is unknown, this is susceptible to false negative labels. **Strategy 1** was used to establish the concept of EvoMIL. We sampled the negative viruses from all viruses that are in different genera than viruses in the positive dataset, with the aim of minimising the false negative viruses in the dataset. **Strategy 2** aimed to make the task progressively more challenging using the fact that as a result of coevolution and cospeciation similar viruses tend to infect similar hosts. Here we selected the negative viruses from those that infect hosts in the same taxonomic rank as the positive host, from phylum to genus, thereby meaning the classifier had to distinguish between more and more similar viruses. For **Strategy 2** the negative samples and positive samples are more likely to share proteins exhibiting structural mimicry [11], so it will be challenging to train classifier models and make the binary models sufficiently sensitive to capture the difference between positive and negative samples. The number of viruses related to each host is shown in S1 Table. The largest prokaryote dataset is *Mycolicibacterium smegmatis* with 838 known viruses, followed by *Escherichia coli* with approximately half the number of viruses. For the eukaryotic datasets, *Homo sapiens* have by far the largest number of known virus species (1321) with the next highest being the tomato (*Solanum lycopersicum*) at 277, see S2 Table. The distribution of the top 10 virus families can be found in S1 Fig. Approximately 60% of viruses associated with prokaryotes belong to the Siphoviridae family (see S1A Fig), whereas the Geminiviridae, Picornaviridae and Papillomaviridae families are the top three families in eukaryotic hosts, each accounting for roughly 18% of viruses associated with eukaryotic hosts (see S1B Fig). Note that viruses associated with eukaryotes are more diverse than those associated with prokaryotes.

## EvoMIL achieves high performance for binary virus-host prediction

Embedding vectors for each of the proteins of a virus generated with the protein language model, ESM1b, were used as an instance in a "virus bag" for MIL. These labelled bags were used to train the MIL model using 5-fold cross-validation on 80% of the datasets, then each 5-fold model performance was evaluated on the remaining 20% of the datasets. Each model is evaluated with a range of metrics: AUC, accuracy, F1 score, sensitivity, specificity, and precision. We evaluated the predictive performance of EvoMIL for binary classification using the datasets generated with both **Strategy 1** and **Strategy 2** above, training a prediction model for each host.

**Prokaryotic and eukaryotic host performance.** The heatmaps of evaluation indices for the prokaryotic and eukaryotic host classifiers are presented in Fig 2A and 2C. Here, evaluation indices are calculated based on the best-performing model with the highest AUC chosen in 5-fold cross-validation. In Fig 2A, the accuracy is higher than 0.9 except for two hosts, which are 0.86. The ROC curves in Fig 2B show that all prokaryotic classifiers perform very strongly with each host achieving an AUC greater than 0.95 and 8 achieving an AUC of 1.

We also obtained the mean and standard deviation of each host, by testing 5-fold cross-validation models on the host test dataset (see S3 Table). EvoMIL shows good performance, with 14/15 hosts achieving a mean AUC greater than 0.9. Overall, our results demonstrated that EvoMIL shows an impressive performance in the binary classification tasks of viruses associated with prokaryotic hosts. More evaluation metrics are included in S3 Table.

The accuracy of each eukaryotic host classifier is shown in Fig 2C, it is clear that all hosts perform with an accuracy higher than 0.8 except for two hosts which are roughly 0.7. *H. sapiens* obtained the highest accuracy with 0.84; *Mus musculus* has the lowest accuracy with 0.69.

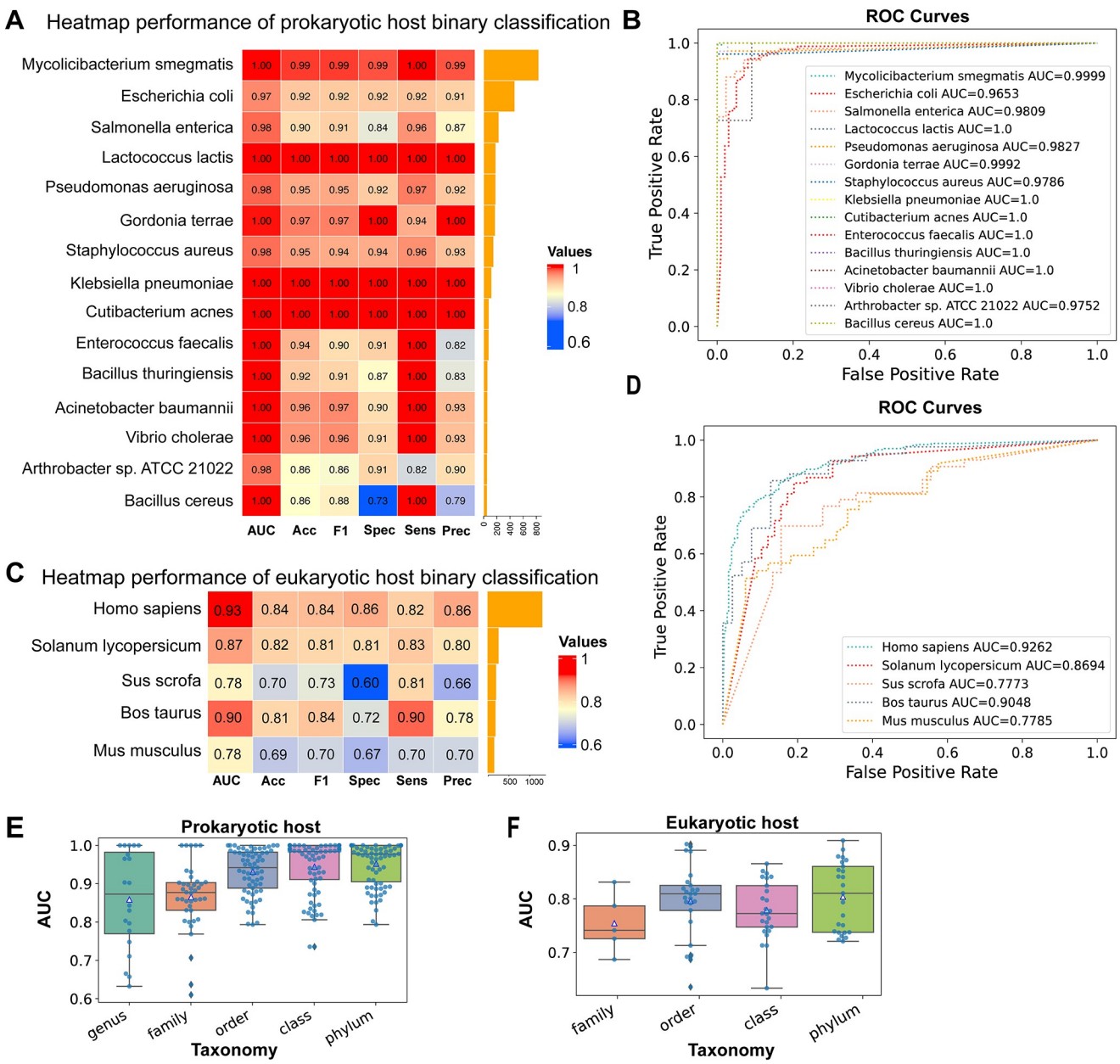

**Fig 2. Performance of binary classification tasks.** This figure separately shows the heatmap of AUC, accuracy, F1 score, sensitivity, specificity, and precision on 15 prokaryotic (A) and 5 eukaryotic host binary classifiers (C), negative samples are selected by **strategy 1**; ROC curves of 15 prokaryotic hosts (B) and 5 eukaryotic hosts (D) corresponding with heatmap plots A and C; AUC values of different taxonomies on prokaryotic (E) and eukaryotic hosts (F) where negative samples are selected using **strategy 2**.

The ROC curves of the 5 eukaryotic hosts classifiers are presented in Fig 2D. Although the eukaryote classifiers achieve good performance with AUCs above 0.77, they perform less well than the prokaryote classifiers, with only 3/5 datasets scoring an AUC above 0.85. There may be several explanations for the lower performance. Firstly, the average number of proteins per virus is much lower, resulting in small "bags" for MIL. Secondly, there is a much higher diversity of virus types in the datasets of the eukaryotic hosts, often containing viruses from multiple

Baltimore classes. The virus of these different classes is polyphyletic meaning they will have no common ancestor and therefore have no shared genes and interact with different host pathways.

The mean and standard deviation of each host are obtained by testing five trained cross-validation models on the test data set (see S4 Table). Here, the mean AUC is higher than 0.85 except for the classifiers of two hosts that perform less well, with AUC scores of 0.761±0.01 for *Sus scrofa* and 0.762±0.02 for *M. musculus*. Overall, our results demonstrate that EvoMIL performs well in binary classification tasks of viruses associated with eukaryotic hosts.

**Sampling negative samples from similar viruses makes binary host classification more challenging.** Next, we test our model with more challenging tasks. Using the second strategy of selecting negative viruses that are associated with hosts sharing the same taxonomic rankings as the hosts associated with positive viruses, we observe that the classification task becomes increasingly challenging as we move from the phylum to the genus level. Results show that our EvoMIL models achieve high AUC scores but that distinguishing between viruses of similar hosts is more difficult with a noticeable drop in performance at family and genus levels. In Fig 2, the box plots show AUC values of prokaryotic (E) and eukaryotic (F) hosts based on negative selection **strategy 2** with five taxonomies genus/family/order/class/phylum. Phylum level (lime colour) presented significant improvement compared with lower taxonomies, especially the genus level. Note, at the lower taxonomic ranks there are only sufficient numbers of negative viruses to meet our threshold of 50 for 4 hosts at the genus level, 8 hosts at the family level and 13 hosts at the order level.

To quantify the difficulty of the task, we computed the sequence similarity scores between all pairs of positive and negative sets on **strategy 2** using MMSeq2 [12] (see S2 and S3 Figs). Most of the scores are above 0.6. Looking at the scores across taxonomy levels, the phylum (purple) level tends to have lower similarities, while the family (orange) and order (green) levels tend to exhibit higher identity scores. These suggest a good degree of sequence similarity between positive and negative viruses, therefore a challenging classification task.

## Embedding features outperform protein and DNA k-mer features on multi-class classification tasks

To demonstrate that the ESM embeddings are encoding more predictive information than conventional features, we generated feature sets from the k-mer composition of the nucleic acid, amino-acid and physio-chemical sequences [13], and evaluate ESM-1b and k-mer features with a multi-class classification task. Here, we performed multi-class classification extending attention-based MIL by modelling the joint multinomial distribution of bag labels. Both prokaryotic and eukaryotic multi-class datasets were constructed using hosts infected by at least 30 viruses. This resulted in 22 classes for the prokaryotes and 36 classes for the eukaryotes. Again, we applied 5-fold cross-validation on training datasets, then separately tested trained models on the testing dataset. These models are named ESM-1b, AA_2, PC_3 and DNA_5, according to different features.

The results for each model are presented in Table 1, we obtain AUC, accuracy, and F1 scores by evaluating each of the 5-fold cross-validation models with the test dataset. Comparing ESM-1b with AA_2, PC_3, and DNA_5 for both prokaryotic and eukaryotic hosts, the ESM-1b features have the strongest prediction signal since, during testing on prokaryotic hosts, the mean F1 score of EvoMIL is 0.88, achieving improvements of 10.8%, 16.2%, and 4.9% compared with AA_2, PC_3 and DNA_5, respectively; while the counterpart of EvoMIL in eukaryotic hosts is 0.292, achieving improvements of 1.7%, 6.6% and 11.5% compared with AA_2, PC_3 and DNA_5 (Table 1). Additionally, ESM-1b demonstrates better performance in

**Table 1. The AUC, Accuracy and F1 score of multi-class MIL by using ESM-1b and k-mer features.**

| Host type | Methods | AUC | Accuracy | F1 score |
|---|---|---|---|---|
| Prokaryotes | ESM-1b | **0.992±0.0** | **0.909±0.0** | **0.88±0.01** |
| | AA_2 | 0.979±0.0 | 0.856±0.01 | 0.794±0.02 |
| | PC_3 | 0.969±0.01 | 0.843±0.01 | 0.757±0.02 |
| | DNA_5 | 0.987±0.0 | 0.882±0.01 | 0.839±0.01 |
| Eukaryotes | ESM-1b | **0.831±0.01** | **0.494±0.01** | **0.292±0.01** |
| | AA_2 | 0.829±0.03 | 0.494±0.01 | 0.287±0.01 |
| | PC_3 | 0.821±0.01 | 0.479±0.01 | 0.274±0.02 |
| | DNA_5 | 0.801±0.02 | 0.466±0.02 | 0.262±0.01 |

Here is a comparison of AUC, accuracy, F1 score between ESM-1b and k-mer feature sets (AA_2, PC_3 and DNA_5). For each feature, the evaluation was conducted by training multi-class classification models on 22 prokaryotic hosts and 36 eukaryotic hosts, and the mean and standard deviation of the evaluation metrics (AUC, accuracy, and F1 score) were obtained using 5-fold cross-validation. The accuracy of a random classifier is 0.045 for 22 classes and 0.028 for 36 classes.

terms of AUC and accuracy compared to k-mers, except for eukaryotes, the accuracy of ESM-1b is comparable to that of AA_2.

Fig 3A and 3B, respectively, show AUC and accuracy across ESM-1b and k-mer features in both prokaryotes and eukaryotes, and AUC and accuracy are equivalent with those presented in Table 1. In Fig 3A, ESM-1b has the highest AUC and smallest standard deviation among prokaryotic hosts; for eukaryotic hosts, ESM-1b shows the smallest standard deviation and the highest mean AUC compared with k-mer features. Here, AA_2 has the largest variation, despite obtaining the highest AUC. In Fig 3B, ESM-1b presents the highest accuracy and the smallest standard deviation among prokaryotic hosts; for eukaryotic hosts, ESM-1b includes the highest accuracy. Additionally, although AA_2 presents the smallest standard deviation in accuracy, ESM-1b obtained the best mean AUC and F1 score (see Table 1).

To further evaluate the MIL models, we test the models using viruses of hosts that are not in the multi-class models trained above, only selecting hosts with between 5 and 30 associated viruses in the VHDB. These viruses were used to test all 5-fold models generated above, comparing all features. Fig 3C and 3D show a comparison of the accuracy of ESM-1b and k-mer features for all cross-validation MIL multi-class models. To calculate accuracy we determined the percentage of correct predictions on all test samples. Here, a correct prediction is defined as one in which the predicted host rank is the same as that of the true host label for the taxonomic ranks of phylum, class, order, family, and genus.

In prokaryotic hosts (Fig 3C), accuracy is roughly between 1% and 10% at genus, family and order levels, while accuracy is between 20% and 90% at class, phylum, and kingdom levels. These results indicate that prediction is more challenging at lower taxonomic ranks. ESM-1b performs best at the genus level with the highest mean accuracy, 1.8%, while the mean accuracy for AA_2, PC_3, DNA_5 are 1.09%, 1.18% and 1.23%, respectively (see S5 Table). Furthermore, across each taxonomy level, the standard deviation of ESM-1b is the smallest compared with k-mer features, although ESM-1b does not perform with the highest accuracy for higher taxonomic ranks. Overall, ESM-1b performs best at the genus level and shows stable accuracy results across all taxonomy levels on prokaryotic hosts.

As for eukaryotic hosts (Fig 3D), accuracy is roughly between 1% and 6% at genus, family and order levels, while accuracy is between 25% and 50% at class, phylum, and kingdom levels. Across all taxonomic ranks, ESM-1b consistently outperforms DNA and protein k-mer features in terms of accuracy. For example, the mean accuracy of ESM-1b at the class level is 36.87%, which is higher by 12.86%, 9.49%, and 11.72% than AA_2, PC_3 and DNA_5,

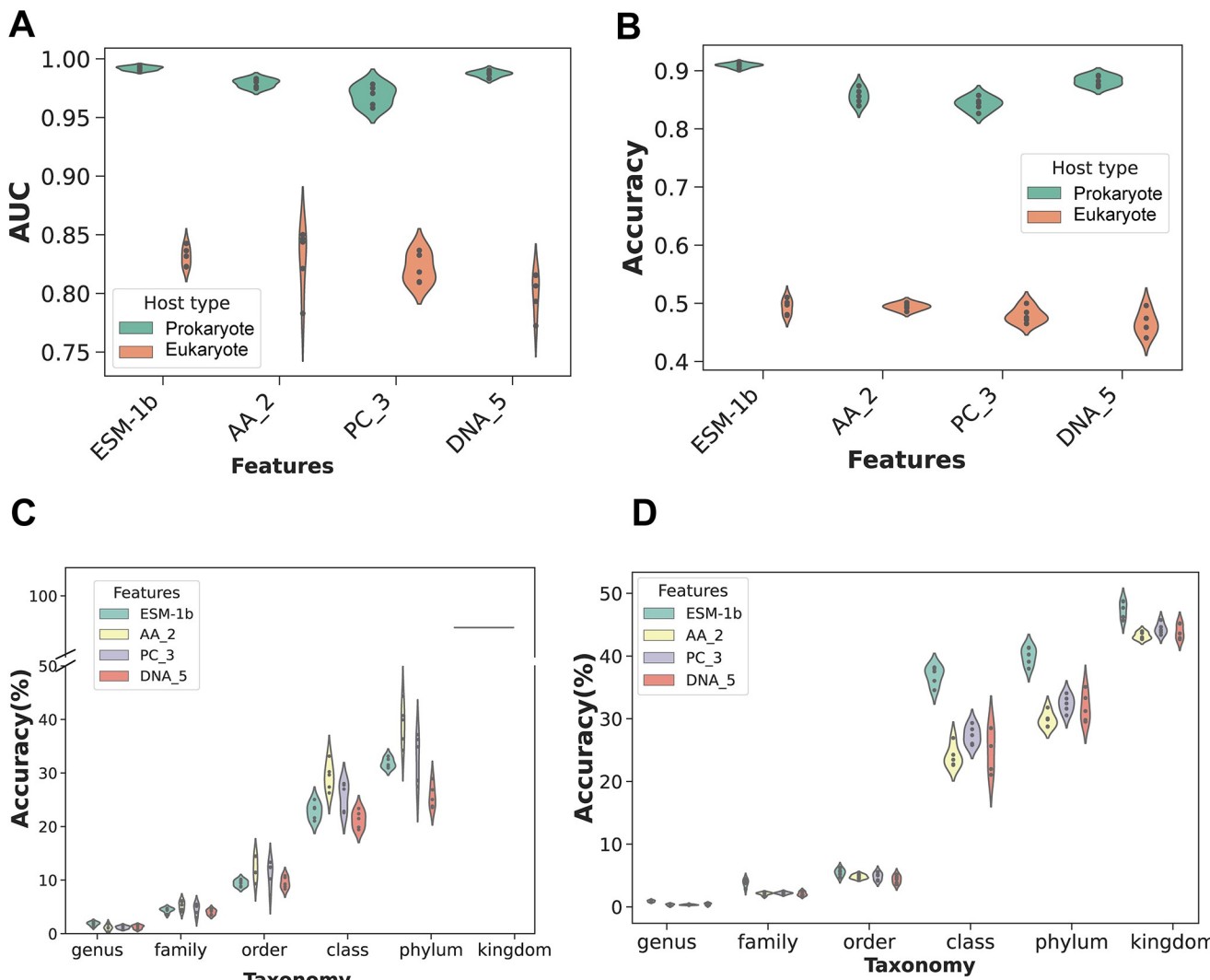

**Fig 3. Performance of multi-class classifications on ESM-1b and k-mer features.** A and B represent the AUC and accuracy, respectively, for prokaryotic and eukaryotic hosts using four feature sets (ESM-1b, AA_2, PC_3 and DNA_5), AUC and accuracy are equivalent with those presented in Table 1. C and D indicate the results obtained by testing the trained models on prokaryotic and eukaryotic hosts associated with 5 to 30 viruses, using the four different feature sets described above.

respectively (see S5 Table). In summary, the multi-class MIL model based on ESM-1b features shows potential in predicting hosts that are associated with fewer than 30 viruses in eukaryotic datasets.

Overall, ESM-1b demonstrated superior performance compared to the k-mer features, through 5-fold cross-validation on both prokaryotic and eukaryotic hosts. Furthermore, we compare the accuracy for each host to evaluate the prediction performance of ESM-1b and k-mer features on multi-class classification tasks.

Fig 4A and 4B, respectively, presented taxonomic tree of prokaryotic and eukaryotic and the mean Log2 accuracy ratio between ESM-1b and K-mers (AA_2, PC_3 and DNA_5) and standard deviation of 5-fold cross-validation for each host. Results show that ESM-1b achieved the highest mean accuracy in 17 out of 22 prokaryotic hosts compared with protein and DNA k-mer features. These findings indicate that ESM-1b outperforms k-mer feature sets in multi-

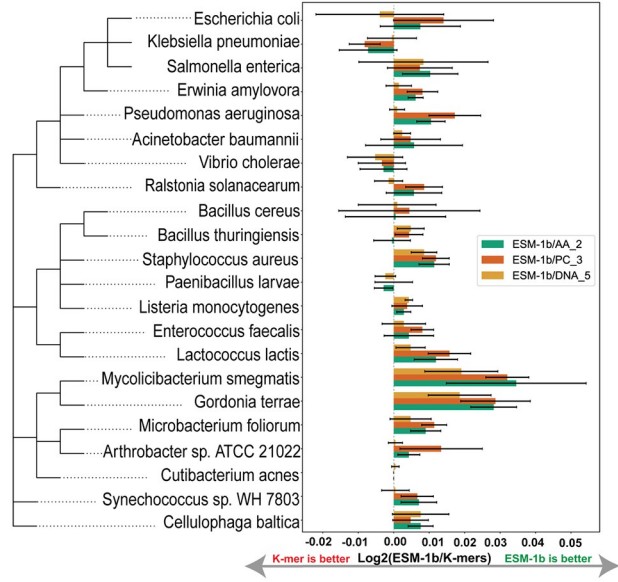

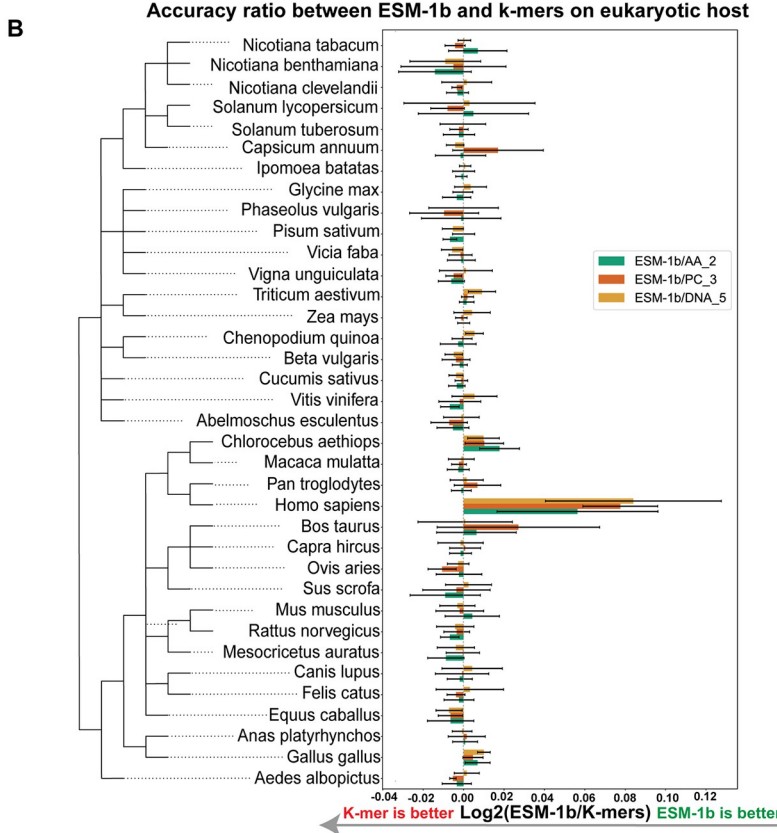

**Fig 4. The taxonomic tree, aligning with Log2 of ratio accuracy between ESM-1b and k-mers.** The figure shows the taxonomic tree of 22 prokaryotic (A) and 36 eukaryotic (B) hosts. Each host is aligned with a bar plot showing the accuracy ratio and standard deviation of 5-fold cross-validation between ESM-1b and AA_2, PC_3, and DNA_5, respectively. The taxonomic tree align with the accuracy between ESM-1b and k-mers is shown in S4 Fig.

classification tasks for prokaryotic hosts. For eukaryotic hosts, the performance of ESM-1b is comparable to k-mer features in each eukaryotic host except *H. sapiens*. Notably, ESM-1b demonstrated a significant improvement over k-mer in *H. sapiens*, which has the largest number of training samples, indicating that EvoMIL performs better with a larger number of virus training samples for eukaryotic hosts.

To understand if the host phylogeny explains the prediction performance of multi-class classification, we visualise the taxonomic tree of hosts next to the heatmap showing the number of predicted false hosts for each host (S5 Fig). The objective is to determine if false predicted host classes of viruses are more likely to share common parent nodes with true hosts in the taxonomic tree. In S5 Fig, two heatmaps respectively present the number of false predicted hosts which belong to the same taxonomy as the true host in prokaryotic and eukaryotic hosts.

In the prokaryotic host heatmap (S5A Fig), the highest number of falsely predicted hosts is observed at the family level. Notably, *E. coli* accounts for the largest number of false predicted hosts. Furthermore, an interesting observation is that *Bacillus cereus* and *Bacillus thuringiensis* host models predict each other as the host label (see Fig 5A), and they belong to the same genus, indicating that predicted hosts have a tendency to be hosts sharing the same taxonomy levels with the true host label. In the eukaryotic host heatmap (S5B Fig), the most number of false predicted hosts belong to the class level. *H. sapiens* and *Bos taurus* have the highest number of false predicted hosts, and they predict each other as the host label (see Fig 5B). Similarly, a similar situation can be observed between *S. scrofa* and *M. musculus*, which further reinforces the finding that closely related hosts within the taxonomic tree are more likely to be predicted as each other's labels.

To understand the relationship between host phylogeny and predicted results, we use *E. coli* as an example in prokaryotes. In Fig 5A, based on ESM-1b, the numbers of false host labels predicted are: 7% for *S. enterica*, 3% labels for *Erwinia amylovora*, 2% labels for *K. pneumoniae*, and 1% label for *Ralstonia solanacearum*, where *S. enterica* and *K. pneumoniae* belong to the same family as *E. coli*, *E. amylovora* belong to the same order, and *R. solanacearum* belongs to the same phylum as *E. coli*. In Fig 4B, it is clear that it is more challenging to predict eukaryotic hosts compared with prokaryotic hosts. ESM-1b demonstrates significantly superior performance compared to k-mer features in *Nicotiana clevelandii* and *Cucumis sativus*, while its performance on other hosts within the eukaryotic taxonomic tree is comparable to that of the k-mer features. In the case of *H. sapiens* as an example, the false positive host labels based on ESM-1b for *H. sapiens* primarily consist of 4% *Pan troglodytes* labels, 3% *Chlorocebus aethiops* labels, 3% *Macaca mulatta* labels, and 3% *S. scrofa* labels. Here, *Pan troglodytes* belongs to the same family as *H. sapiens*; *Chlorocebus aethiops* and *Macaca mulatta* belong to the same order level as *H. sapiens*; and *S. scrofa* belongs to the same class level as *H. sapiens*.

Overall, viruses are more likely to be misclassified as closely related hosts during multi-class classification, confirming that those hosts sharing common parent nodes in the taxonomic tree tend to be infected by similar viruses [13].

## Benchmarking EvoMIL with prokaryotic host predictors

Here we compare EvoMIL's prediction performance with 9 prokaryotic host predictors including the state-of-the-art iPHoP [14], BLASTn, BLASTp, CRISPR, WisH [15], VirHost-Matcher [16], PHP [17], SpacePHARER [18], VirHostMatcher-Net [19] and vHULK [20]. We chose 364 viruses across the 22 prokaryote species that matched the host labels in our training set, which were chosen from the benchmarking test dataset from the iPHoP paper. The full list of the testing viruses can be found in S7 Table. For details about prediction settings for each approach, please see the Materials and methods section.

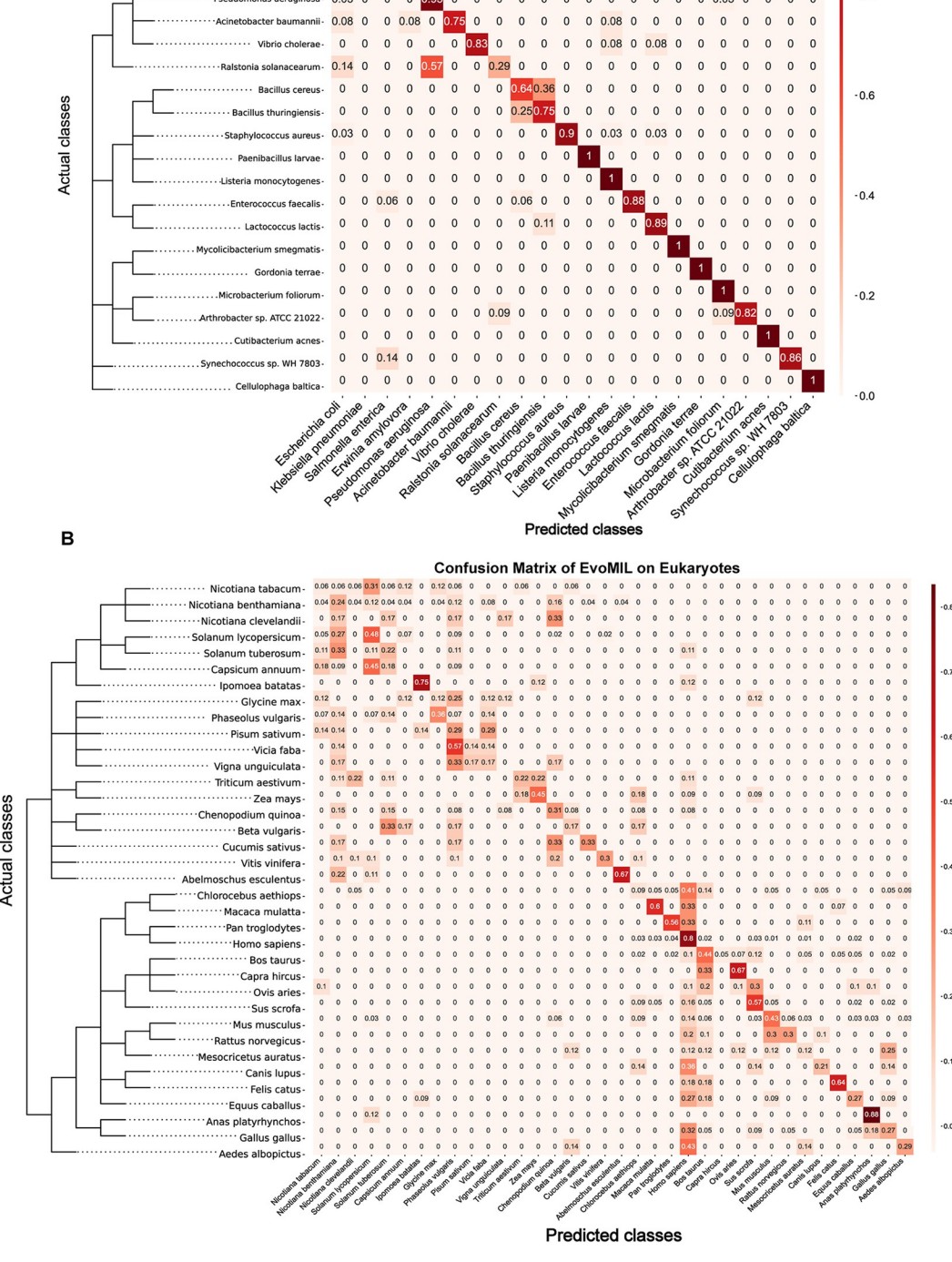

**Fig 5. The Confusion matrix plot of prokaryotic hosts (A) and eukaryotic hosts (B) based on EvoMIL.** The confusion matrix plots A and B represent the performance of the EvoMIL model on 22 prokaryotic hosts and 36 eukaryotic hosts, respectively. It is constructed by evaluating the model's predictions on a test set comprising 20% of the dataset, while the EvoMIL model was trained on the remaining 80% of the data. This plot provides insights into the model's accuracy in predicting the host species for the tested viruses.

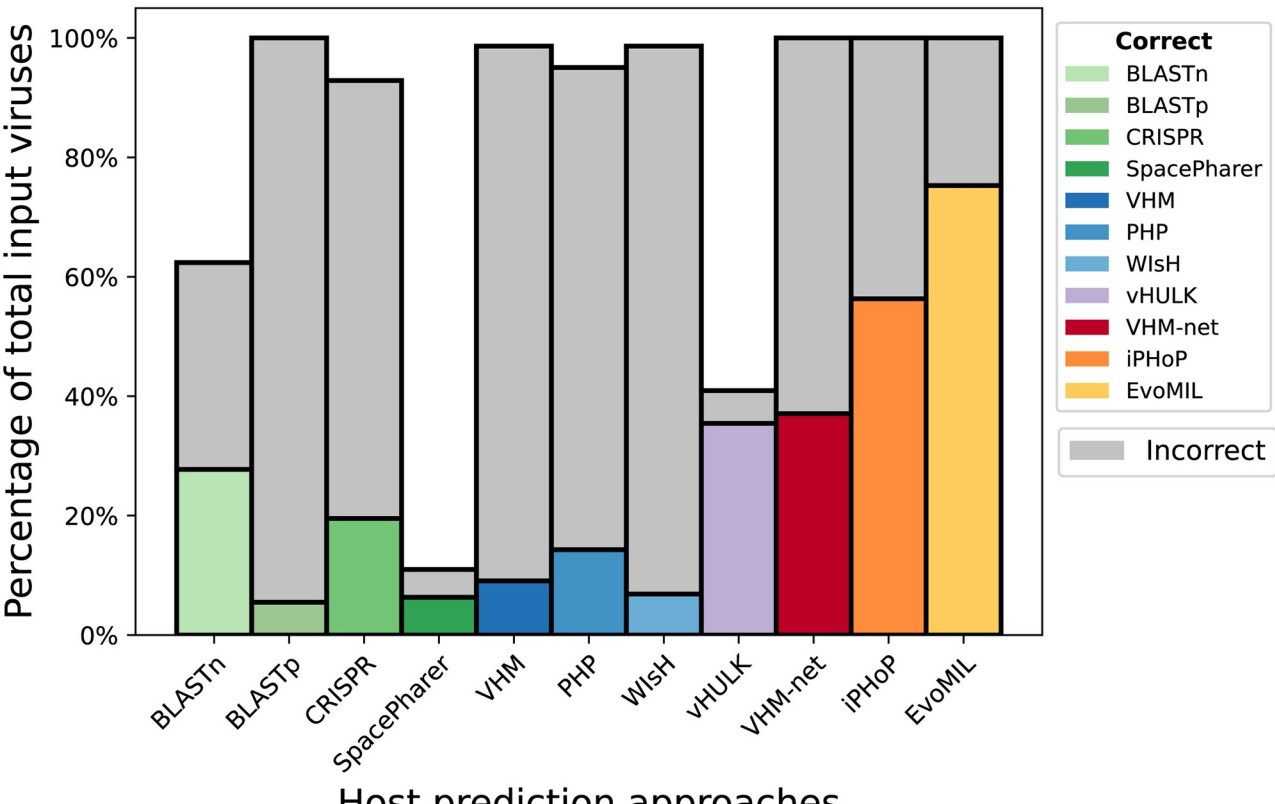

**Fig 6. Comparison of EvoMIL and other host prediction approaches on an independent test dataset.** The y-axis presents the number of correct predictions (coloured bar) and the number of incorrect predictions (grey bar) for each tool (x-axis) on the chosen benchmarking test dataset (S7 Table). This plot shows the percentage of correct and incorrect host species predictions on the test dataset, and prediction source results are available in S8 Table.

BLASTn, BLASTp, CRISPR, PHP, WisH, VirHostMatcher and iPHoP make multiple host predictions for a given virus. We considered any predicted host species in their prediction list that matched the true host species as a correct prediction. EvoMIL, SpacePharer, VHM-Net, and vHULK are evaluated based on their top-1 prediction accuracies. As shown in Fig 6, Evo-MIL achieves the highest accuracy of 75.27% on the test viruses, with 274 correct predictions out of 364 viruses and has a 33.65% improvement over the state-of-the-art approach, iPHoP.

### MIL attention weights can be used to interpret which virus proteins are important

Next, we investigate if the proteins identified as important for the prediction are key contributors to virus-host specificity, i.e., whether the transformer embeddings are encoding biologically meaningful information about the viral proteins. As we described in the introduction, attention-based MIL learns the weight of each protein in a virus bag. A high weight indicates that a protein is important for host prediction, and by implication is more likely to be important to virus-host specificity. We looked at Gene Ontology (GO) annotations of these highly weighted proteins and their proximity in the embedding space with hierarchical clustering.

In Fig 7, we can see that the top 5 ranked proteins annotated by GO terms include important proteins with high weights, and proteins from a virus might contain weights of 0, so

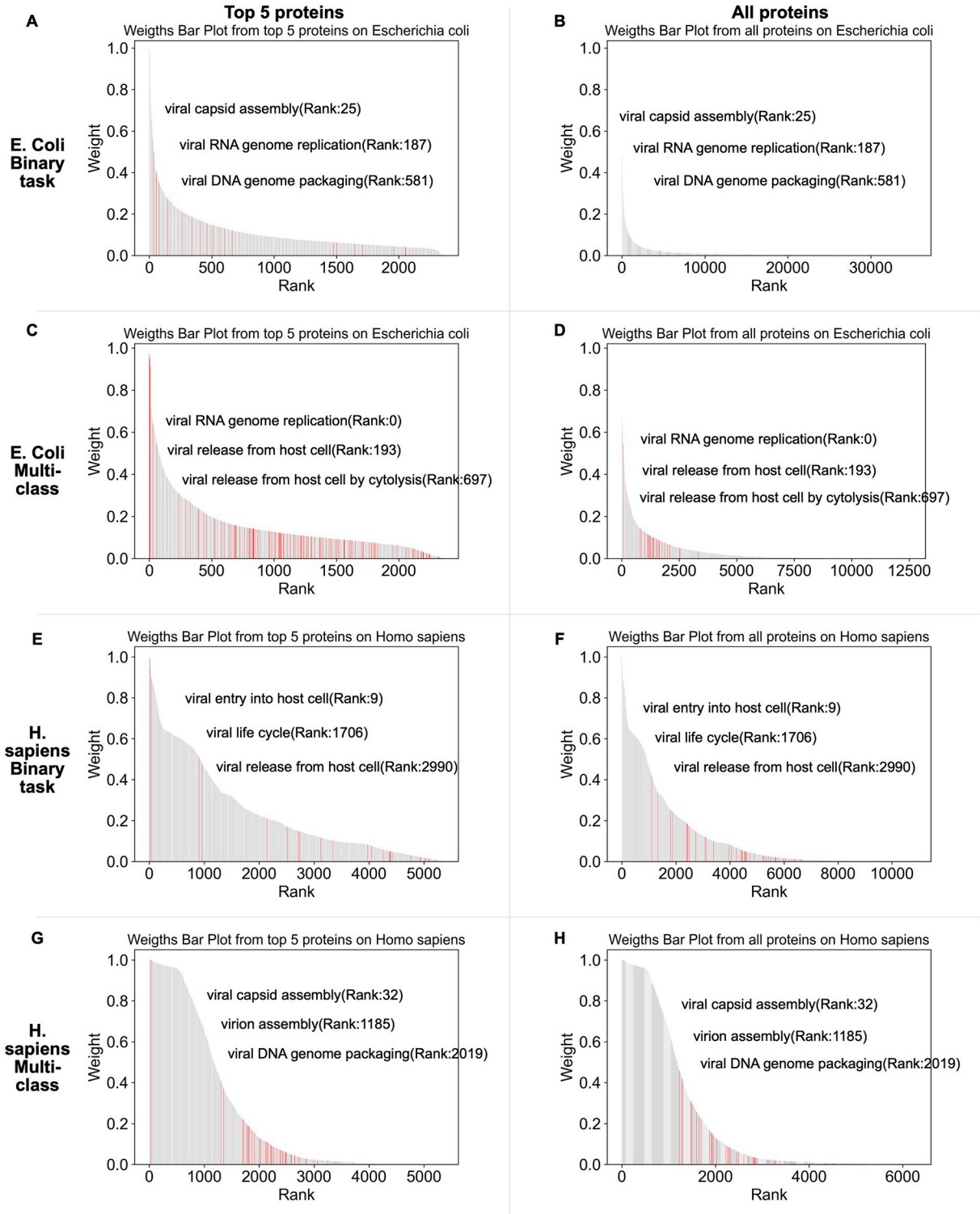

**Fig 7. The bar plots display the ranking of weights for the top 5 proteins and all proteins of viruses associated with *E. coli* and *H. sapiens*, respectively.** The top four bar plots illustrate protein weights obtained for *E. coli* based on binary classification models(A, B) and multi-class classification models (C, D), respectively. Similarly, the bottom four bar plots depict the protein weights obtained for *H. sapiens* based on binary classification models(E, F) and multi-class classification models (G, H), respectively. Each host consists of two sections: the left subplot shows the top 5 ranked protein weights, while the right subplot displays all protein weights sorted in descending order.

selecting the top 5 ranked proteins allows us to collect proteins with GO annotations and high weights, as the aim is to present key proteins which are assigned high weights based on binary and multi-class models. We mark the GO annotation for proteins and select three GO annotations as examples. For example, the top 5 ranked proteins (Fig 7A) and all ranked viral proteins (Fig 7B) associated with *E. coli* shared the same GO annotation (viral capsid assembly) and have the identical rank index within their respective ranked protein sets, meaning that selecting the top 5 ranked proteins of each virus still allows us to identify key GO annotations.

As shown in Fig 7, this selection allows us to focus on proteins that are assigned high weights, considering the possibility that some proteins from a given virus may have weights of 0. By choosing the top 5 ranked proteins, we are able to gather proteins with GO annotations that also possess high weights. This approach allows us to highlight key proteins assigned high weights based on binary and multi-class models. Fig 7A and 7B demonstrate that some ranks of GO annotations for the top 5 ranked proteins aligned with those ranks of all proteins, indicating that GO annotations can still be obtained even when considering only the top 5 ranked proteins.

Here we highlight results from viruses infecting *E. coli* and *H. sapiens*. These viruses contain the most GO annotations, because they are the most extensively studied hosts from prokaryotic and eukaryotic categories. Using the learned model parameters of the best-performing cross-validation model in the binary and multi-class classification tasks respectively, we ranked the attention weights of the proteins for the positive viruses and selected the top 5 ranked proteins from each virus. Furthermore, we collect two groups of top protein embeddings based on binary (Fig 8A and 8C) and multi-class classification (Fig 8B and 8D) models, respectively. These proteins were annotated with functions related to the viral life cycle using GO terms obtained from InterProScan [21].

For the set of top-ranked proteins of the viruses associated with *E. coli* based on the binary model, roughly 1.5% (35 out of 2359) were assigned a viral life cycle GO term. In Fig 8A, there

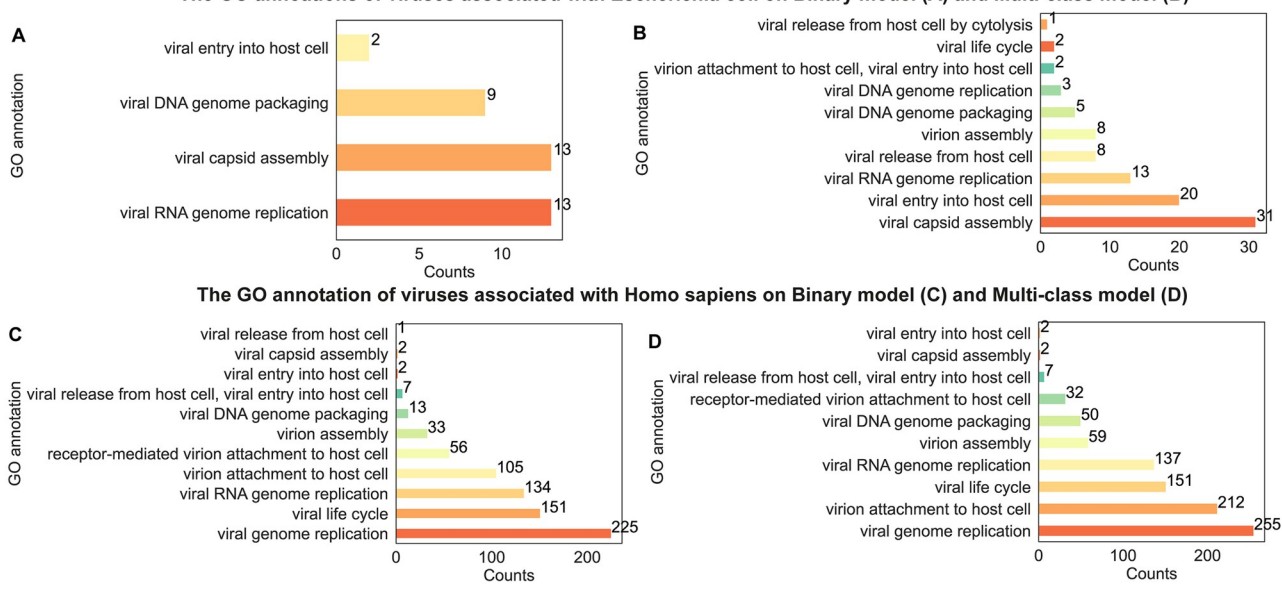

**Fig 8. The bar plot of GO annotations of viral viruses associated with *E. coli* (top) and *H. sapiens* (bottom).** Panels show the number of each GO annotation of the top 5 ranked proteins for each virus associated with *E. coli* (A, B) and *H. sapiens* (C, D). Here, the protein weights in A and C are obtained by binary models, whereas in B and D, the weights are obtained by multi-class classification models.

are four different GO terms related to the viral life cycle including viral capsid assembly (GO:0019069), viral DNA genome packaging (GO:0019073), viral RNA genome replication (GO:0039694), and viral entry into host cell (GO:0046718). In Fig 8B, ten different GO annotations of virus proteins associated with *E. coli* based on the multi-class model, revealing a greater diversity of viral GO terms, approximately 3.7% (87 out of 2359) were assigned a viral life cycle GO term. Viral capsid assembly (GO:0019069), viral entry into host cell (GO:0046718), viral RNA genome replication (GO:0039694) are the top 3 GO annotations.

The top-ranked viral protein associated with *H. sapiens* displays significantly more GO annotations (Fig 8C and 8D). In the binary case (Fig 8C), we found 23.9% (1280 out of 5357), in the binary case, and 12.6% (676 out of 5357), in the multi-class case, of the top ranked proteins to have GO terms related to viral life cycle. It is worth noting that binary (Fig 8C) and multi-class (Fig 8D) shared most of the GO terms, and viral RNA genome replication (GO:0039694), attachment to host cell, virion attachment to host cell (GO:0019062) and viral life cycle (GO: 0019058) are the most prevalent.

Furthermore, we used hierarchical clustering [22] to test if different proteins with the same functional annotation have similarities in the embedding space. In S6 Fig we clustered the ESM-1b embeddings of the abovementioned top-ranked proteins with viral life cycle GO annotations. We found apparent clusters across the viral proteins for *E. coli* (S6A and S6B Fig) and *H. sapiens* (S6C and S6D Fig). These results suggest ESM-1b embeddings encode functional signals, and MIL learns some consistent features associated with host specificity. While having clusters of proteins with the same GO annotation, we also observe that not all proteins with the same GO annotation are clustered into a single cluster. This suggests that there is richer information contained in the embedding space beyond existing function annotations. Whether this information is biologically meaningful warrants a future study.

## EvoMIL identifies key proteins of SARS-CoV-2

To highlight EvoMIL performance on an unseen virus, we looked at the prediction and attention weights of SARS-CoV-2 [23] using the trained *H. sapiens* binary classifier from previous section. Wuhan-Hu-1 with NCBI Reference Sequence NC_045512.2, which is not included in the training, was predicted to be a human virus with a probability of 0.97 compared to the SVM-kmer binary classifiers [13] of less than 0.01. The top three ranked proteins contributing to host prediction are spike sub-sequences, Non-structural proteins sub-sequences and Nucleocapsid protein, with weights of 0.298, 0.221 and 0.189, respectively. The spike protein contains the receptor binding site which is responsible for cell entry. Nucleocapsid protein is an RNA binding protein that plays a critical role at many stages of the viral life cycle making direct interactions with many host proteins [24]. More N protein and host protein interactions can be retrieved from protein-protein interaction databases in UniPort (https://www.uniprot.org/uniprotkb/P0DTC9/entry#interaction). The viral process GO terms of these top-ranked proteins are "fusion of virus membrane with the host plasma membrane" (GO:0019064), "fusion of virus membrane with host endosome membrane" (GO:0039654), "receptor-mediated virion attachment to host cell" (GO:0046813) and "endocytosis involved in viral entry into host cell" (GO:0075509). It has been reported that these GO terms are associated with viral infections, indicating that protein attention weights obtained by EvoMIL perform well in identifying important proteins contributing to virus-host associations.

## Discussion

In this paper, we introduce EvoMIL, a novel method for virus-host prediction. Inspired by the success of NLP approaches in biology, we demonstrate the power of using a protein

transformer language model (ESM-1b) to generate embeddings of viral proteins that are highly effective features for virus-host classification tasks. ESM-1b is capturing meaningful biological/host information from viral proteins, despite the ESM-1b training dataset being mainly comprised of proteins from cellular life with only 1% being viral proteins. We demonstrate that attention-based MIL can identify which of a virus's proteins are most important for host prediction, and by implication which proteins may be key to virus-host specificity.

Our classification results show that EvoMIL is able to predict the host at the species level with high AUC, accuracy and F1 scores. The prokaryote binary classifiers achieved a mean AUC of 0.992 whereas the eukaryotic classifiers achieved a mean AUC of 0.851 during selecting negative samples by **Strategy 1**. We also evaluate the model's performance using different levels of negative sampling from **Strategy 2**, our findings indicate that training binary classifiers becomes more challenging when those hosts associated with negative and positive samples are more closely related. Additionally, results demonstrated that eukaryote host prediction is a more difficult task for two reasons: viruses associated with eukaryotic hosts are much more diverse across all seven Baltimore classes compared to the prokaryotic viruses which are mainly double-stranded DNA viruses; secondly, the eukaryote datasets contain many RNA viruses which have far fewer proteins, this makes it more challenging for MIL which needs many instances in each bag to perform well. Table 1 shows that ESM-1b features outperform more conventional sequence composition-based features on multi-class classification tasks. Furthermore, the clusters seen in the *E. coli* and *H. sapiens* hierarchical clustering dendrogram (S6 Fig) indicates that embeddings are able to capture functionally related information of the high-ranked proteins associated with host specificity. Moreover, EvoMIL is able to find SARS-CoV-2 spike proteins from top-ranked proteins obtained by attention weights of the model.

Transformers enable a one-step pipeline to generate dense feature vectors from viral genomes and learn multiple layers of information about the structural and functional properties of proteins in an unbiased way. Multiple-instance learning is able to capitalize on any host signal contained within individual protein sequences, circumnavigating the need to represent each virus with a single feature vector. This enables it to find common patterns/signals across bags of proteins that differ greatly in both size and content. An additional advantage of attention-based MIL is the ability to identify which proteins are important in prediction, the identification of important proteins of viruses that contribute to host prediction is essential in understanding the specific viral proteins that play a critical role in host infection. Further analysis is needed to test whether this can be exploited to uncover the mechanisms behind virus-host specificity.

In this paper we took a virus-based approach in which only information from the viral genomes is used to generate features, this limits us to classifying viruses for those hosts that have a minimum number of known viruses. Also, by limiting the viral sequences to only Refseq sequences to reduce redundancy, we have further narrowed the range of hosts we can predict. Additionally, we acknowledge that Virus-host DB is biased towards well-studied organisms such as humans, to mitigate this, combining different virus-host association databases could enhance the diversity and comprehensiveness of the training samples. The ultimate host predictor would be able to make predictions for hosts which have no or few known viruses, to do this we will include host information. Combining virus and host-based approaches has been shown to greatly increase the host range of a prediction tool whilst maintaining a low false discovery rate, see Roux et al.[14] and Wang et al. [19]. In the future, we will make use of the much larger numbers of host-annotated virus genomes available in public databases and include host information to construct a model that can make predictions using associations for any virus-host pair.

## Materials and methods

### Data

The current study uses datasets collected from the Virus-Host database (VHDB), (https://www.genome.jp/virushostdb/) [10]. The VHDB contains a manually curated set of known species-level virus-host associations collated from a variety of sources, including public databases such as RefSeq, GenBank, UniProt, and ViralZone and evidence from the literature surveys. For each known interaction, this database provides NCBI taxonomic ID for the virus and host species and the Refseq IDs for the virus genomes. We downloaded datasets on 20-10-2021, along with the associated FASTA files containing the raw viral genomes and the FAA file with translated CDS for each viral protein. At this point, the VHDB contained 17733 associations between 12650 viruses and 3740 hosts that were used to construct binary datasets for both prokaryotic and eukaryotic hosts.

### Constructing balanced binary dataset

In this study, we retrieve the reference genome from Refseq genome [25] for each virus. The aim is to reduce the amount of redundant or similar sequences in the datasets. We obtain known virus-host associations for prokaryotic and eukaryotic hosts from the Virus-Host database VHDB [10], and remove viruses whose protein sequences do not exist in the Refseq database.

Balanced binary datasets were constructed for each host with an equal number of positive and negative virus sets to obtain a balanced data set for binary classification tasks. For both the prokaryotic and eukaryotic datasets, we collected 4696 associations between 4696 viruses and 498 prokaryotic hosts at the species level; 9595 positive associations from 9595 viruses and 1665 eukaryotic hosts at the species level.

For each binary virus-host association data set, viruses can be represented by $V = \{V_1, V_2, .. V_V\}$, and the host is represented by $H$. The positive labels consisted of known associations from VHDB [10], and the set of positive viruses can be represented by $V_{pos} = \{V_1, V_2, ..V_P\}$, where $P = \frac{V}{2}$. Most phages are likely to infect a range of hosts belonging to the same taxonomy [26, 27], therefore, to mitigate possible errors, some researchers [28, 29] construct negative virus-host associations by selecting those viruses from the remaining viruses that do not infect the given host, instead of relating to hosts with different taxa from the given host. Using this method, we construct putative negative virus-host associations by collecting viruses from a specific range. As for a given host $H$ in the prokaryotic datasets, all viruses are represented by $V_{pro} = \{V_1, V_2, ..V_T\}$, where T is the total number of viruses associated with prokaryotic hosts, we collect negative viruses $V_{neg} = V_{pro} - V_{pos}$, which are not related with the given host; and host species taxonomy of $V_{neg}$ are different from host $H$. On top of that, in order to get the balanced datasets for model training and testing, the size of datasets $V_{neg}$ is equal to $V_{pos}$, we randomly choose viruses from putative negative virus set $V_{neg}$, finally we get the same number of positive and negative virus datasets, that can be represented by $V_{neg} = \{V_{P+1}, V_{P+2}, ..V_V\}$. Similarly, we can create a balanced binary dataset for each eukaryotic host. The number of viruses related to the given host is at least 50 and 125 for the prokaryotic and eukaryotic datasets. In addition, there exist some segmented and non-segmented viruses; segmented viruses have multiple RefSeq sequences and need to be combined to represent complete viral sequences. After setting the threshold for the number of viruses at the species level, combining segmented viruses and removing redundancy non-segmented viruses, we have 15 prokaryotic datasets and 5 eukaryotic datasets.

## Feature extraction

**ESM-1b.**   The Pre-trained ESM-1b model is used to transform protein sequences into fixed-length embedding vectors that are used as features for downstream tasks such as binary and multi-class classification. For n protein sequences $(h_1, \ldots, h_n)$ input into ESM-1b [7], we obtain n embedding vectors $\mathbf{X} = (\mathbf{X_1}, \ldots \mathbf{X_n})$, each with a dimension of 1280. In the pre-training process, the representation is projected to log probabilities $(y_1, \ldots, y_n)$, the model posterior of amino acid at position $i$ is represented by a softmax over $y_i$, the output embeddings $(h_1, \ldots, h_n)$ are applied in downstream tasks, (the parameters are "–repr_layers 33 –include mean per_-tok"). Here, we adapt virus embedding vectors to the supervised attention-based MIL learning tasks (EvoMIL), and the input feature dimension of ESM-1b is 1280 generated by esm1b_t33_650M_UR50S. EvoMIL is based on a fully connected layer with an input size of 1280 (ESM-1b embedding size) and an output size 800, and then using a linear classifier and the output size is the number of prediction classes.

There are two special cases of sequences that need to be pre-processed to be suitable as input for ESM-1b:

1. In NCBI, some protein sequences include the amino acid J, which is used to refer to unresolved leucine (L) or isoleucine (I) residues. However, the ESM-1b model does not include the token 'J' and they must be removed before processing. In order to remove J from NCBI sequences, we randomly replace J with either Leucine(L) or Isoleucine(I).

2. The ESM-1b model can only process sequences with a maximum of 1024 tokens including the beginning-of-sequence (BOS) token and the ending-of-sequence (EOS) token, meaning that the maximum length of the protein sequence is 1022 amino acids. The parameter '–truncation' can be used to crop a longer sequence, but this will result in a loss of some part of the sequence information. In order to include information from the entire protein sequence, we split longer sequences and process the sections simultaneously. For protein sequences longer than 1022, we split the sequence into lengths of 1022 amino acids. If the final section is shorter than 25 amino acids, it is discarded as it is considered too short to include meaningful information. So that for a given protein sequence $H$, of length $len(H)$:

   if $len(H) < 1022 + 25$ then truncate the sequence

   else if $len(H) \geq 1022 + 25$ split the sequence.
   For example, if we have a sequence of length 2049 ($2 \times 1022 + 25$), it will be cut into two sub-sequences with the length of 1022, and one sub-sequence with the length of 25. Resulting in three embedding vectors for the single protein sequence that are all assigned to the same label as the parent protein and will be considered instances in the attention-based MIL.

**K-mers.**   K-mer composition vectors were generated for each of the CDS regions of a virus and used as an alignment-free representation of a sequence. A k-mer is a sub-sequence of length k and is generated for each position in a sequence. These features are obtained by calculating the frequency of each of the possible k-mers in a sequence. To minimise the effect of sequence length, the resulting vector is normalized so that its sum is equal to 1. Here, we extract k-mer features from sequences corresponding to DNA, amino acids (AA) and their physio-chemical properties (PC) [13]. DNA and amino acid sequences of the CDS regions were downloaded from the NCBI database, and used directly to extract AA and DNA k-mers. To extract PC k-mers from protein sequences we first re-label each amino acid as one of seven

groups based on its physio-chemical properties, as described by Shen et al. [30]: AGV, C, FILP, MSTY, HNQW, DE, and KR, and then extract k-mers using the seven group labels as the alphabet. To generate k-mer composition vectors of reasonable length for MIL we use k-mer lengths of 5, 2 and 3, respectively, for the DNA, AA and PC sequences. This results in DNA_5, AA_2, PC_3 feature sets of dimensions of 1024, 400, and 343, respectively.

## Attention-based multiple instance learning

We represent a virus as a set of protein embedding instances $X = \{\mathbf{x}_1, \mathbf{x}_2, \mathbf{x}_3, \ldots, \mathbf{x}_M\}$. In a binary classification setting, the label of the virus is $Y \in \{0, 1\}$. If $Y = 1$, the virus is known to be associated with the host, otherwise, the virus is not associated with the host, but the instance label $\{y_1, y_2, y_3, \ldots, y_M\}$ is unknown. Multiple instance learning (MIL) is used to predict a label for a bag with a set of instances, so it was used to predict a host label for a set of proteins from a virus. The label $Y$ can be represented as follows:

$$Y = \begin{cases} 0, & \text{iff} \sum_m y_m = 0, \\ 1, & \text{otherwise.} \end{cases} \tag{1}$$

The MIL model can be interpreted as a probabilistic model in which the label of the bag is distributed according to a Bernoulli distribution with the parameter $\theta(X) \in \{0, 1\}$, which is the probability of a virus being associated with a host, i.e., Y = 1.

$$p(Y|X) = \theta(X)^Y (1 - \theta(X))^{1-Y}, \tag{2}$$

The model can be trained by maximising the Bernoulli likelihood function, which is equivalent to minimising the negative entropy function.

Given a bag of instance $X$, the scoring function $\theta(X)$ can be written as Eq (3) [31]:

$$\theta(X) = g(b^T \sum_{x_m \in X} a_m[\varphi(\mathbf{x}_m)]), \tag{3}$$

where $g$ is the sigmoid function. $\varphi$ is a neural network that transforms protein embedding to a lower dimensional space. $\varphi$ is implemented as a fully connected layer with ReLU activation functions. It has an input size matching embedding dimension from ESM-1b and k-mers, and an output size universally set to 800. $b \in \mathbb{R}^{800 \times 1}$ is the weight of the binary classifier.

$a_m$ are the attention weights of instance $m$ computed as

$$a_m = \frac{\exp\{\mathbf{w}^\top \tanh(\mathbf{V}\mathbf{u}_m^\top)\}}{\sum_{j=1}^{M} \exp\{\mathbf{w}^\top \tanh(\mathbf{V}\mathbf{u}_j^\top)\}}, \tag{4}$$

where $\mathbf{u}_m = \varphi(\mathbf{x}_m)$. $V \in \mathbb{R}^{L \times M}$ and $\mathbf{w} \in \mathbb{R}^{L \times 1}$ are learnable parameters. tanh(.) is the hyperbolic tangent function. We used these weights to quantify the importance of proteins to host prediction.

In the multi-class classification case, $g$ is softmax function and $b \in \mathbb{R}^{800 \times K - 1}$, with $K$ being the number of hosts.

## Gene Ontology

The Gene Ontology (GO) is used to describe gene function from different organisms, which includes three GO domains: biological process, cellular component and molecular function [32]. GO annotations are links between gene products and GO terms, and the structure of GO can be represented by a graph, each node is a GO term and nodes are connected by edges.

InterProScan [21] is a tool to obtain GO terms by classifying query proteins to protein families within InterPro's database. To analyze the protein functions of important virus proteins identified by EvoMIL, we used InterProScan to obtain GO annotations of viruses.

## Taxonomic tree

We used the NCBI interface https://www.ncbi.nlm.nih.gov/Taxonomy/CommonTree/wwwcmt.cgi to generate taxonomic trees for 22 prokaryotes and 36 eukaryotes when given taxonomy IDs, and then applied the R package ggtree [33] to obtain the trees in Figs 4 and 5 and S5 Fig.

## Experimental settings of prokaryotic host prediction approaches in benchmarking

We used iPHoP v1.3.3 (Nov 2023 version, [14]) and recommended database Sept_21_pub_rw for the test virus genomes. BLASTn and CRISPR predictions were based on BLASTn (v2.12.0+) between the test virus genomes and the iPHoP_db_Sept21 BLAST and spacer database, respectively. BLASTp (v2.12.0+) was used to calculate the identity between the test virus and the prokaryotic host protein sequences from our training set, retrieved from NCBI. A maximum of 200 protein sequences per host were downloaded, and the host with the top-ranking match was selected as the predicted host. SpacePHARER predictions were obtained by running the 'predictmatch' function from SpacePHARER v5.c2e680a [18] with default parameters. For WIsH (v1.0, [15]) predictions, virus genomes were compared to the iPHoP_db_Sept21 WIsH database with a maximum p-value of 0.2. For PHP (July 2021 version, [17]), using iPHoP_db_Sept21 PHP database to predict hosts for virus genomes. VirHostMatcher (Apr 2018 version, [16]) was used to get the s2* similarity score between each pair of virus and host similarities. For the prediction of VirHostMatcher-Net, we used VirHostMatcher-Net(July 2021 version, [19]) with complete genome mode and default parameters. vHULK (v1.0.0, [20]) host species predictions were obtained by using default parameters.

## Evaluation

To evaluate our classifiers we use six evaluation metrics, including AUC, accuracy, F1 score, sensitivity, specificity, and precision as defined below. The two main evaluation metrics used in our analysis and explanation are AUC and accuracy. The area under the receiver operating characteristic curve (AUC) is used to evaluate machine learning performance; accuracy is the ratio of the number of correctly predicted samples to the total number of samples.

True positive (TP), true negative (TN), false positive (FP), and false negative (FN) are the parameters used to calculate specificity, sensitivity, and accuracy. True positive(TP): predicted label and known label are positive. True Negative(TN): the predicted label and known label are negative. False Positive (FP): The predicted label is positive, but the actual label is negative. False Negative(FN): The predicted label is negative, but the actual label is positive. Sensitivity is the percentage of positive samples; specificity is the percentage of negative samples. The F1 score is the harmonic mean of precision and recall. Here, we set average='macro' to calculate the F1 score and precision for each label and get their unweighted mean. The formula of the evaluation indices is as follows:

$$Accuracy = \frac{TP + TN}{TP + TN + FP + FN}, \tag{5}$$

$$Sensitivity = \frac{TP}{TP + FN}, \tag{6}$$

$$Specificity = \frac{TN}{TN + FP}, \tag{7}$$

$$Precision = \frac{TP}{TP + FP}, \tag{8}$$

$$Recall = \frac{TP}{TP + FN}, \tag{9}$$

$$F1 = \frac{2 \times Precision \times Recall}{Precision + Recall}, \tag{10}$$

## Supporting information

**S1 Table. Prokaryotic hosts: Hostname, the number of viruses associated with the host.** The table of 22 prokaryotic datasets shows the host species name, the number of viruses associated with each host, and the average, minimal and maximal number of protein sequences of viruses associated with each host.
(XLSX)

**S2 Table. Eukaryotic hosts: hostname and the number of viruses associated with the host.** The table of 36 eukaryotic datasets shows the host species name, the number of viruses associated with each host, and the average, minimal and maximal number of protein sequences of viruses associated with each host.
(XLSX)

**S3 Table. Results of binary classifiers in prokaryotic hosts.** Evaluation indices are obtained by testing 5-fold cross-validation models on each host, and then the mean and standard deviation of the evaluation metrics can be obtained. Evaluation metrics include AUC, accuracy, f1, specificity, sensitivity, and precision.
(XLSX)

**S4 Table. Results of binary classifiers in eukaryotic hosts.** Evaluation indices are obtained by testing 5-fold cross-validation models on each host, and then the mean and standard deviation of the evaluation metrics can be obtained. Evaluation metrics include AUC, accuracy, f1, specificity, sensitivity, and precision.
(XLSX)

**S5 Table. The accuracy (%) of multi-class MIL by using ESM-1b and k-mer features on tiny host datasets.** Comparison of mean accuracy and standard deviation between ESM-1b and k-mer feature sets: ESM-1b, AA_2, PC_3 and DNA_5. For each feature, training multi-class classification models on 22 prokaryotic hosts and 36 eukaryotic hosts based on 5-fold cross-validation, then mean and standard deviation of accuracy is obtained by testing the trained model on host datasets that are associated with fewer viruses, ranging from 5 to 30 (tiny host datasets).
(XLSX)

**S6 Table. Table of GO ID and GO terms.** The GO terms are coloured in the hierarchical clustering dendrograms, we retrieve protein functions for each GO ID, and the GO terms are shown in the table.
(XLSX)

**S7 Table. Table of test virus samples used for benchmarking of EvoMIL on prokaryotic host species prediction.** This table includes the test viruses used to compare EvoMIL and recent prokaryotic host species prediction approaches.
(XLSX)

**S8 Table. Table of benchmarking results of the test viruses on prokaryotic host species prediction.** This table includes prediction results of prokaryotic host prediction methods on the test dataset provided in S7 Table.
(XLSX)

**S1 Fig. Virus taxonomy (family) distribution on prokaryotic and eukaryotic hosts.** To illustrate the virus taxonomy distribution, we plot pie charts to show the distribution of virus families in prokaryotic and eukaryotic hosts. Viruses associated with prokaryotes are dominated by Siphoviridae family, which constitutes approximately 60% (A), whereas the Geminiviridae, Picornaviridae and Papillomaviridae families are the top three ranked families, each accounting for roughly 18% (B).
(TIF)

**S2 Fig. The genome sequence similarities between positive and negative viruses from different taxonomies on each prokaryotic host.** This figure presents the distribution of genome similarities between positive and negative samples on each prokaryotic host, where the negative viruses are chosen based on the same taxonomy (genus, family, order, class, phylum) as the positive viruses.
(TIF)

**S3 Fig. The genome sequence similarities between positive and negative viruses from different taxonomies on each eukaryotic host.** This figure presents the distribution of genome similarities between positive and negative samples on each eukaryotic host, where the negative viruses are chosen based on the same taxonomy (genus, family, order, class, phylum) as the positive viruses.
(TIF)

**S4 Fig. The taxonomic tree, aligning with accuracy values between ESM-1b and k-mers.** The figure shows the taxonomic tree of 22 prokaryotic hosts (A) and 36 eukaryotic (B) hosts. Each host is aligned with a bar plot showing the accuracy and standard deviation of 5-fold cross-validation between ESM-1b and AA_2, PC_3, and DNA_5, respectively.
(TIF)

**S5 Fig. The figure represents heatmaps of different features for each prokaryotic (A) and eukaryotic (B) host.** The values in the heatmap are the total number of predicted hosts which belong to the same taxonomy as the true host.
(TIF)

**S6 Fig. The dendrogram of protein embeddings of viruses associated with *E. coli* (top) and *H. sapiens* (bottom).** Figures show the hierarchical clustering dendrograms of the top 5 ranked protein embeddings for each virus associated with *E. coli* (A, B) and *H. sapiens* (C, D). Here, the protein weights in A and C are obtained by binary models, whereas in B and D, the weights are obtained by multi-class classification models. Protein GO terms regarding the viral

life cycle of viruses are highlighted with different colours, allowing us to understand the predictive signals of proteins captured by the pre-trained transformer model.
(TIF)

## Author Contributions

**Conceptualization:** Dan Liu, Francesca Young, David L. Robertson, Ke Yuan.

**Data curation:** Dan Liu, Francesca Young.

**Formal analysis:** Dan Liu, Francesca Young, Kieran D. Lamb, David L. Robertson, Ke Yuan.

**Funding acquisition:** David L. Robertson, Ke Yuan.

**Investigation:** Dan Liu, Francesca Young, David L. Robertson, Ke Yuan.

**Methodology:** Dan Liu, Francesca Young, Kieran D. Lamb, David L. Robertson, Ke Yuan.

**Project administration:** Dan Liu, David L. Robertson, Ke Yuan.

**Resources:** Dan Liu, David L. Robertson, Ke Yuan.

**Software:** Dan Liu, Francesca Young, Kieran D. Lamb.

**Supervision:** Francesca Young, David L. Robertson, Ke Yuan.

**Validation:** Dan Liu, Francesca Young, Kieran D. Lamb, David L. Robertson, Ke Yuan.

**Visualization:** Dan Liu, Francesca Young, Kieran D. Lamb, David L. Robertson, Ke Yuan.

**Writing – original draft:** Dan Liu, Francesca Young, David L. Robertson, Ke Yuan.

**Writing – review & editing:** Dan Liu, Francesca Young, Kieran D. Lamb, David L. Robertson, Ke Yuan.

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
