## [Decision Letter · Decision Letter 0]

9 Apr 2024

Dear Dr Yuan,

Thank you very much for submitting your manuscript "Prediction of virus-host associations using protein language models and multiple instance learning" for consideration at PLOS Computational Biology.

As with all papers reviewed by the journal, your manuscript was reviewed by members of the editorial board and by several independent reviewers. In light of the reviews (below this email), we would like to invite the resubmission of a significantly-revised version that takes into account the reviewers' comments.

We cannot make any decision about publication until we have seen the revised manuscript and your response to the reviewers' comments. Your revised manuscript is also likely to be sent to reviewers for further evaluation.

Sincerely,

Fuhai Li

Academic Editor

PLOS Computational Biology

Rob De Boer

Section Editor

PLOS Computational Biology

Reviewer's Responses to Questions

**Comments to the Authors:**

Reviewer #1: The authors propose a method named EvoMIL for both prokaryotic and eukaryotic virus host prediction. By integrating a pre-trained protein embedding module and a well-designed weighing module (attention-based multiple instance learning), EvoMIL is expected to leverage different proteins in the virus genomes. The idea of the EvoMIL is interesting and the results show that EvoMIL performs good on the dataset created by the author. However, there are still some major concerns about the work, especially the experimental design.

Major concerns:

1. Virus host prediction is a hot research topic, and numerous tools have been developed to address this problem. However, this manuscript lacks benchmark experiments that demonstrate the competitiveness of EvoMIL in comparison to state-of-the-art methods. As a newly designed method, the author should cite these existing tools and provide a comprehensive comparison. A review of many tools was provided here: https://doi.org/10.1093/bioinformatics/btac239 Some of these tools probably have better/newer versions. Although these only focus on prokaryotic viruses, the authors can still compare the performance of this part.

2. In addition to the existing tools, the authors also discuss the potential use of alignment-based methods for host prediction. Considering that EvoMIL aims to assess the significance of viral proteins in host prediction, I suggest that the authors include BLASTp results as a fundamental benchmark method in their experiments. For instance, in the binary classification task, it would be valuable to examine whether BLASTp can yield better alignment results for positive pairs compared to negative pairs. This would provide insights into the performance of EvoMIL.

3. Since novel viruses are typically reconstructed from metagenomic data, it is possible that the viral sequences obtained may not be as complete as those downloaded from curated databases. Consequently, certain proteins may be missing from the assembled viral contigs. As EvoMIL is a "protein-based" method, I am wondering whether it can still perform effectively on short contigs.

4. As described in the manuscript, it is noted that EvoMIL is currently designed to handle host prediction for a specific set of 15 prokaryotes and 5 eukaryotes and cannot perform well on other microbes. This means that if the dataset includes viruses that do not infect these specific microbes, EvoMIL may not be able to provide accurate predictions. To address this limitation, I would suggest that the authors consider introducing a class labeled as 'Other' to accommodate users who may wish to utilize EvoMIL with datasets containing viruses that infect organisms outside of the predefined set. This would allow for more flexibility and usability of EvoMIL in a broader range of scenarios.

5. The creation of the negative set:

To discuss more comprehensively, the authors provided 2 strategies to create the negative set: 1) The hosts of positive viruses and negatives viruses should not be in the same genus. 2) The natural hosts of positive viruses and negatives viruses are in the same taxonomic group (from genus to phylum). The author claims that the strategy 2 introduces challenging cases to evaluate the tool. It is partially correct. My concern is that two viruses respectively infecting rodent and human (both order Mammalia) can be very different. If the similarity between them is very low, the case is then not that challenging.

In a typical bioinformatic scenario, creating the dataset by the sequence’s similarity is preferred. A visualization of the similarity between the positive set and negative set could help mitigate the concern.

6. Number of the eukaryotic hosts.

In the first experiment (binary classification), the authors only created 5 eukaryotic datasets, because they require that there are at least 125 viruses in the positive set. The cutoff is a bit stringent. How will this method perform when the cutoff is 50 as the prokaryotic one? When the tool is designed on 36 eukaryotes, why not show them all?

Other detailed comments can be found below.

(1) The data distribution

Compared to the emergent novel viruses, the VHDB database is not complete. Although using it as a reference is acceptable, the authors should show the virus taxonomy distribution to the audience.

(2) Model structure

Although the attention deep MIL module is from an existing work, the authors still need to elucidate the detail of the module. In Fig 1, the structures of neural network and the final classifier are not mentioned.

The learning architecture of AA_2, PC_4, and DNA_5 is not stated.

(3) Problem formulation

In the second experiment (multi-class task), is the data multi-label? Or with single label? This is not stated. The definition of “multi-class classification” and “multi-label classification” is different. While the former only outputs one label, the latter might output multiple labels for one query. And a number of viruses have the ability to infect multiple hosts. Without stating this, the problem formulation is not finished.

(4) Fig 3 and Table 5

The caption of Fig.3 is not clear. Which experiment does it belong to? It seems that Fig 3 (a, c) and Table 1 are collectively for 22 prokaryotes and 36 eukaryotes. And Fig 3 (b, d) is for small dataset. The organization is interleaved. It is hard to follow the logic.

(5) Number of the prokaryotic hosts.

The author only included 22 prokaryotes, which is rather limited.

(6) The accuracy of multi-classification.

The tool is finally complied as a multi-class classifier among 36 eukaryotic hosts. And according to table 1, the accuracy on test set is around 49.4%. This is not a satisfactory result. I suppose the alignment-based method like BLASTn can achieve this as well.

(7) Fig 5 and Fig S2

In Fig 5, is true label at the left axis? This is not stated.

While Fig 5 shows the error cases in each host group and different rank by heatmap, Fig S2 uses the confusion matrix directly. They are of a bit redundancy. I recommend the authors use Fig S2 in the main part instead of Fig 5. Because Fig 5 does not reflect the error trend.

In Fig S2, most of the area is white, because a strong signal is at Homo-Homo (212). To better show the result, I strongly recommend the author conduct row normalization by the actual classes.

(8) Balanced binary dataset

In the part of the balanced binary dataset, the author defined 4 vectors, V, Vpos, Vneg, and Vpro. V is of P length, and Vpro is of T length. T is not defined, so that Vpro is confusing as well.

In 450, there is a typo “Using this method”.

When we look at the virus world from an individual host, like human, most of the viruses in the world do not infect human. So, I think it is better to include all the negative data, instead of just using a small part of it.

(9) The UMAP section

The authors want to know “whether the embeddings of these important proteins identified by attention-based MIL contain any underlying clustering structures.” But the clustering structures can only show the successful application of ESM-1b embedding without any further benefit to the host part. As long as the proteins number is large enough, there definitely will be clusters in UMAP. However, ESM-1b is not contributed by this work. So, I don’t understand why they put this here.

In 324th row, please try to explain the difference.

Reviewer #2: >>> Overall, ESM-1b demonstrated superior performance compared to the k-mer features, through 5-fold cross-validation on both prokaryotic and eukaryotic hosts. Furthermore, we split multi-class datasets into 80% training and 20% test sets, then train and test our model without cross-validation, and compare the accuracy for each host to evaluate the prediction performance of ESM-1b and k-mer features on multi-class classification.

I don’t understand why this would be desirable if you already have the results of a 5-fold cross validation. I’d remove this part and make Figure 4 be based on the results from the cross validation (i.e., have the log2(esm-kmer) values be computed for each fold separately and then provide standard deviation markers on the barplots. Also, for this figure please provide guidelines or better spacing between each set of bars to make it easier to read which set of bars correspond to each species

>>> “Similarly, ESM-1b outperforms the other feature sets in 16 out of 36 eukaryotes.”

Does this mean that your model failed to outperform a very simple baseline in most of the eukaryote datasets?

>>> Please make the methodological details of how the features for AA 2, PC 3 and DNA 5 were created for reproducibility.

>>> It would be nice if Figure 5 had a taxonomy tree similarly to Figure 4

>>> Please make Figure 6 have a white background to save ink if someone ever prints this.

>>> “Again, the UMAP plots demonstrate that different protein functions are clustered into separate locations, while GO terms with similar functions tend to form sub-clusters.”

I disagree with this statement, it seems that there are many proteins with the same GO term that are separated in different clusters (e.g. “Virion Assembly” in subfigures A,B and “Viral Assembly” subfigures C,D). If you want to make this claim please run a clustering algorithm (k means will suffice, but a hierarchical clustering algorithm would give you better working space to make claims) and see to which degree certain functions cluster or do not cluster together.

>>> “For example, the indices of GO annotations of the top 5 ranked proteins (Fig 7 A) are the same as the indices of all proteins (Fig 7 B), meaning that we can obtain GO annotations although only selecting top 5 ranked proteins.”

This sentence is a bit confusing, although one can understand it after referring to the image. It would be good if you could make a bit more time to make this sentence clearer.

Reviewer #3: In this paper the authors present an efficient model able to carry out the virus-host association starting from the proteins set of the virus.

This is a novel method and all the presented data and analysis support the claims.

Some more details are missing about some theoretical concepts, methods or technical issues.

The code and the trained models are available on a GitHub link on the paper and the original data are collected from the Virus-Host database. Nevertheless, the descriptions of the algorithm, the dataset and the used architectures are not satisfactory in the paper for the sake of reproducibility.

The paper has been written with a good and clear english but some concepts are hardly accessible for non-specialists.

In conclusion I suggest to extend some part of the paper in order to make it more accessible to non-specialists and also make the experiments easily reproducible.

You can find below some questions that could be useful for this task:

- Can you explain in a more clear and detailed way the so called "strategy 1" and "strategy 2" ?

- What's the aim of your binary and multi-class classification models?

- Can you explain and describe a little more the K-mer features ?

- Can you add a few line about UMAP, Gene Ontology, GO annotation and InterProScan ?

- Can you describe the used deep learning architectures and the input data dimensions in order to make possible the reproducibility of your experiments ?

**Have the authors made all data and (if applicable) computational code underlying the findings in their manuscript fully available?**

Reviewer #1: None

Reviewer #2: Yes

Reviewer #3: Yes

PLOS authors have the option to publish the peer review history of their article (what does this mean?). If published, this will include your full peer review and any attached files.

Reviewer #1: No

Reviewer #2: No

Reviewer #3: **Yes: **Daniele Baggi
---

## [Decision Letter · Decision Letter 1]

9 Sep 2024

Dear Dr Yuan,

Thank you very much for submitting your manuscript "Prediction of virus-host associations using protein language models and multiple instance learning" for consideration at PLOS Computational Biology.

As with all papers reviewed by the journal, your manuscript was reviewed by members of the editorial board and by several independent reviewers. In light of the reviews (below this email), we would like to invite the resubmission of a revised version that takes into account the reviewers' comments.

We cannot make any decision about publication until we have seen the revised manuscript and your response to the reviewers' comments. Your revised manuscript is also likely to be sent to reviewers for further evaluation.

Sincerely,

Fuhai Li

Academic Editor

PLOS Computational Biology

Rob De Boer

Section Editor

PLOS Computational Biology

Reviewer's Responses to Questions

**Comments to the Authors:**

Reviewer #1: The authors have addressed some of my comments, which I believe has improved the manuscript. However, I still think the authors are able to add BLAST (or BLASTp) as a benchmark method. This is important to examine the difficulty of the problem. Although the authors mentioned there is no published method for this, it is quite feasible. First, if the contigs contain multiple proteins, a simple majority vote can be applied. Authors can choose the variants of the majority vote strategy for this. Second, the authors can use BLASTn and do not need to rely on proteins. Nevertheless, this experiment does not need very complicated design. Instead, it will provide useful insights.

Reviewer #2: The authors seem to have addressed most of my points about the paper. Here are a few more suggestions and a typo:

there is no explicit definition of dsDNA

predictiton -> prediction

Figure 4 is better now, it would also be good to have a non-ratio version in the supplementary.

"Here, we extract k-mer features from sequences corresponding to DNA, amino acids(AA) and their physio-chemical properties (PC) [13]. (...) To extract PC k-mers from protein sequences we first re-label each amino acid as one of seven groups based on its physio-chemical properties: ({AGV}, {C}, {FILP}, {MSTY}, {HNQW}, {DE}, and {KR}), [30]" Although you cite [13] above as the citation for all the three characteristics, you only cite 30 at the end of the second sentence highlighted, it'd be good to have a more explicit citation mentioning that the physiochemical property grouping is from [30]. Not necessary, but would aid clarity.

[13] doi:10.1371/journal.pcbi.1007894

[30] doi:10.1073/pnas.0607879104

**Have the authors made all data and (if applicable) computational code underlying the findings in their manuscript fully available?**

Reviewer #1: None

Reviewer #2: Yes

PLOS authors have the option to publish the peer review history of their article (what does this mean?). If published, this will include your full peer review and any attached files.

Reviewer #1: No

Reviewer #2: No
---

## [Decision Letter · Decision Letter 2]

28 Oct 2024

Dear Dr Yuan,

We are pleased to inform you that your manuscript 'Prediction of virus-host associations using protein language models and multiple instance learning' has been provisionally accepted for publication in PLOS Computational Biology.

Best regards,

Fuhai Li

Academic Editor

PLOS Computational Biology

Rob De Boer

Section Editor

PLOS Computational Biology

Feilim Mac Gabhann

Editor-in-Chief

PLOS Computational Biology

Jason Papin

Editor-in-Chief

PLOS Computational Biology

Reviewer's Responses to Questions

**Comments to the Authors:**

Reviewer #1: No further comments.

**Have the authors made all data and (if applicable) computational code underlying the findings in their manuscript fully available?**

Reviewer #1: None

PLOS authors have the option to publish the peer review history of their article (what does this mean?). If published, this will include your full peer review and any attached files.

Reviewer #1: No

---

## [Editor Report · Acceptance letter]

3 Nov 2024

PCOMPBIOL-D-23-00938R2 

Prediction of virus-host associations using protein language models and multiple instance learning

Dear Dr Yuan,

I am pleased to inform you that your manuscript has been formally accepted for publication in PLOS Computational Biology. Your manuscript is now with our production department and you will be notified of the publication date in due course.

With kind regards,

Zsofia Freund
